ecology

*Sparus aurata*, reversibility, predator-induced morphological defences, geometric morphometrics, predator–prey dynamics

**Author for correspondence:**
Ignacio A. Catalán
e-mail: ignacio@imedea.uib-csic.es

# Reversible morphological changes in a juvenile marine fish after exposure to predatory alarm cues

Carlos Díaz-Gil[1,2,3], Josep Alós[2],
Pablo Arechavala-Lopez[2], Miquel Palmer[2],
Inmaculada Riera-Batle[1,2], Amalia Grau[1,3]
and Ignacio A. Catalán[2]

[1]Laboratori d'Investigacions Marines i Aqüicultura, LIMIA (Balearic Government), C/Eng. Gabriel Roca 69, 07157, Port d'Andratx, Illes Balears, Spain
[2]Instituto Mediterráneo de Estudios Avanzados, IMEDEA (CSIC-UIB), C/Miquel Marqués 21, 07190, Esporles, Illes Balears, Spain
[3]Instituto de Investigaciones Agroambientales y de Economía del Agua, INAGEA (INIA, Govern Balear-UIB), Carretera de Valdemossa km 7.5, 07122, Palma, Illes Balears, Spain

CD-G, 0000-0003-4095-1559; JA, 0000-0003-4385-9539;
IAC, 0000-0002-6496-9182

Chemical cues from predators induce a range of predator-induced morphological defences (PIMDs) observed across fish taxa. However, the mechanisms, consistency, direction and adaptive value of PIMDs are still poorly studied. Here, we have tested if predatory cues can induce changes in the body shape of the juvenile marine fish *Sparus aurata* reared under controlled conditions without the presence of predators by exposing individuals to the olfactory stimulus of a fish predator. We tested our hypothesis using a nested replicated before-after-control-impact experiment, including recovery (potential reversibility) after the cessation of the predator stimulus. Differences in the size-independent body shape were explored using landmark-based geometric morphometrics and revealed that, on average, individuals exposed to a predatory cue presented deeper bodies and longer caudal regions, according to our adaptive theoretical predictions. These average plastic responses were reversible after withdrawal of the stimulus and individuals returned to average body shapes. We, therefore, provide evidence supporting innate reversible PIMDs in marine naive fish reared under controlled conditions. The effects at the individual level, including fitness and the associated applied implications, deserve further research.

# 1. Introduction

Phenotypic plasticity can be understood as those changes in behavioural, morphological and physiological traits responding to a specific environment that increment the individual's fitness and therefore their adaptation to the new conditions [1]. Specifically, plasticity in morphological traits (e.g. changes in shape or number/length of spines) may be induced as a defence mechanism adopted by prey across different taxa to avoid/reduce predation [2,3], through a range of predator-induced morphological defences (PIMDs). In fishes, the most commonly described PIMD is the increase of body depth, which has been reported, for instance, in freshwater species such as crucian carp (*Carassius carassius*) [4–6] and Eurasian perch (*Perca fluvialitis*) [7]. The ecological and adaptive values of these PIMDs has been interpreted in terms of decreasing mortality due to predation by selecting for deeper-bodied individuals that would avoid the predator's maximum mouth gape [3]. Moreover, fish species present other PIMDs in order to protect themselves from predation including enlarged body components of the external morphology, such as the defensive spines in the pumpkinseed sunfish (*Lepomis gibbosus*) [8] and the increment in the number of bony lateral plates and spines in different species of the Gasterosteidae family such as the three-spined stickleback (*Gasterosteus aculeatus*) [9] and the nine-spined stickeback (*Pungitius pungitius*) [10]. Overall, there is substantial evidence of phenotypic plasticity in morphological traits through different PIMDs expressed under different predatory pressure despite the partial heritability of these morphological adaptations [11].

PIMDs can be caused by the piscivorous predators' cues [12] but also by the conspecific alarm cues [13–15] released into the water when a predator attacks and ruptures the epidermal cells of the prey [6,16], among other stressors [2–3]. Moreover, morphological changes in fish can be also produced by environmental stressors such as hypoxia events [17,18] or human actions on fish populations through selecting certain morphotypes while fishing [19,20] or through aquaculture practices (e.g. selection of breeding lines for a better production under stocking conditions). However, many basic questions remain unsolved on the PIMDs. Besides the partial knowledge on the mechanisms of action and adaptive value of PIMDs, a central unresolved question has to do with the reversibility of PIMDs [3] and the associated trade-offs in terms of energetic costs, which may cause maladaptations later on in their life cycle [1] altering their reproductive fitness or lifespan [21].

In addition, the study of PIMDs in marine fish species is scarce, as most of the literature is restricted to freshwater species, although some experimental work has been done, especially on Gasterosteidae [10,22]. As a model organism, we selected the juvenile stage of a temperate fish species of interest for fisheries and aquaculture, the gilthead sea bream (*Sparus aurata*), and considered individuals that had never been exposed to predators. Fish reared in captivity do not have experience with predators and, given the high growth rates of some species and their high plasticity [23], are ideal model organisms to measure the costs, in terms of affecting growth, and reversibility of PIMDs. Moreover, improving our understanding of PIMDs would broaden our understanding of long-standing problems spanning from predator–prey dynamics [24] to evolutionary reasons of morphological plasticity in fishes [21,25] or even potential application of human-induced PIMDs to restocking of fish in the wild [26]. In this study, we tested for the ability to experimentally induce and then reverse PIMDs in captive predator-naive groups of individuals of *S. aurata*.

# 2. Material and methods

A total of 500 juveniles of 55 days post-hatching (dph) *Sparus aurata*, raised under identical conditions, were transported to the laboratory, stocked in six 100 l tanks at 70 individuals per tank ($N = 420$) and acclimatized for two weeks at controlled conditions in an open flow-through system. The remaining 80 individuals were kept in a separate tank (under equal conditions) until the preparation of the chemical cue used in the experiments (see electronic supplementary material for more details on the methods). The experiment began by randomly defining three tanks as control and the other three as treatment tanks with no difference in fish size distribution.

The predatory chemical cue for the treatment group was obtained by combining water where conspecifics that were exposed to predation had been dwelling and an extract of the conspecifics' (naive to predators) dead skin, following the protocol presented in previous experiments [14]. We selected the black scorpionfish *Scorpaena porcus* as a natural predator of Sparidae fishes such as *S. aurata* (see diet in [27]). Twenty predators were captured and fed daily with five live *S. aurata*. After 10 days, the water flow was interrupted and 20 *S. aurata* juveniles were introduced into the aquarium as feed. After 24 h, 60 doses (150 ml) of water dwelled by predator and prey were collected, filtered and topped with 50 ml

of a homogenate of *S. aurata* skin (10 g) from euthanized individuals (see electronic supplementary material). Predatory cue water samples were frozen (−20°C), and another 60 ice cubes (200 ml) made up of filtered seawater that were used as controls.

The experiment was divided into two phases (months): during the first month, one ice cube (control or predator water) was incorporated into each tank from Monday to Friday under no-flow conditions for 4 h. During the second month, no chemical cues nor controls were added. According to a before-after-control-impact design, the tanks were sampled three times: right before the experiment, one month after being exposed to the chemical cue and at the end of the second month after being 30 days without the cue exposure. At each sampling time, 10 fish per tank were randomly sampled and euthanized. Total length (TL, mm) and eviscerated weight (g) were measured individually after a photograph was taken. Fulton condition index ($K = 100$ weight length$^{-3}$), eviscerated weight and length were compared along the experiment using linear mixed effect model (LMM) as detailed in the electronic supplementary material. Body shape analysis based on 10 homologous landmarks per fish (electronic supplementary material, figure S1) was conducted separately for each sampling time, after processing data to achieve allometry-free shapes (electronic supplementary material). Finally, the mean size-adjusted residuals (i.e. allometry-free body shapes) of the body shapes from a Procrustes MANOVA were derived for the control and treatment mean individuals and represented showing the vector displacements between the reference (control) and the target specimens (treatment). These vectors were magnified ×20 to improve the visualization of the encountered differences.

## 3. Results

*S. aurata* juvenile's treatment groups presented similar growth patterns in terms of length, weight and Fulton's K along the experiment (table 1; figure 1a–c). No differences due to either the treatment or the experimental tanks within-treatment (table 1, figure 1a–c) were detected.

The Procrustes MANOVA showed that allometry was present, as the overall body shape was correlated with the fish size represented in the model as the natural logarithm of the centroid size (Csize) of the body shape at the three sampling times (table 2). There was a significant effect on the body shape after one month of being exposed to the chemical cue treatment, reflected mainly in the dorsoventral and tail regions (table 2; figure 1e). Thus, the size-standardized shape of exposed fish became deeper and the peduncle tip entrained a larger portion into the tail. After one extra month without stimulus, the juvenile fish shape returned to the same state displayed by control fish (table 2, figure 1e).

## 4. Discussion

Phenotypic plasticity in life history of prey as a response to predatory cues has been widely reported across taxa [3]. In this work, we did not find any effect of predatory and conspecific alarm cues on the growth-related life-history traits of juvenile naive *S. aurata* along the duration of the study as revealed by the comparable total lengths, weights and/or condition index values. We observed a common ontogenetic change in the overall body shape consisting in a progressive relative widening of the body depth that has been previously described [28]. In addition to this allometric pattern, we have observed that fish exposed to predatory and conspecific alarm cues displayed deeper bodies and elongated peduncles after only one month. This induced change disappeared within just one more month of recovery from the treatment. It should be noted, however, that the concept of reversibility within the current context of strong allometric growth is restricted to the concept of convergence to the control shape, not necessarily to a second change in shape after the withdrawal of the stimulus. Therefore, we provide evidence for PIMDs in reared *S. aurata* juveniles that have never been exposed to predators, reinforcing our hypothesis of expression and reversibility of PIMDs in a marine organism.

There are several examples of PIMDs induced by predatory or conspecific alarm cues in fish, mostly in freshwater species. Examples of these include: incrementing the number and size of anti-predator morphological defences [8,9]; differentiations in the body shape that include larger caudal peduncle region [29,30]; enhanced body depth [4,5,7]; or a combination of the latter two strategies [31]. In our study, the body shape differences resulted from a combination of increased body depth (landmarks 3 and 10 figure 1e) and elongation of tail insertion muscle (landmark 6, figure 1e). These PIMDs have been associated with ecological and fitness benefits like higher manoeuvrability in front of predators that favours survival [32], enhanced thrust and faster startle response [29,33] or simply shift towards a

**Table 1.** Results of the linear mixed effect models for total length, weight (log-transformed) and Fulton condition index (K). Interaction model fitted using 'treatment' effect (control/treatment) and sampling date (1st, 2nd and 3rd) as fixed effects. Est.: estimate value; s.e. standard error of the estimate; Pr(>|t|); $p$-values via Kenward–Roger approximation. In italics the significant $p$-values. The six different tanks were used as random effects of the model. $\tau_{00,\,tank}$ is the between-tanks variance and $\sigma^2$ is the within-each-tank variance (residuals).

| | total length (mm) | | | log (weight) | | | Fulton's K | | |
|---|---|---|---|---|---|---|---|---|---|
| | Est. | s.e. | Pr(>|t|) | Est. | s.e. | Pr(>|t|) | Est. | s.e. | Pr(>|t|) |
| fixed effects | | | | | | | | | |
| (intercept) | 18.7 | 1.402 | *<0.001* | −2.214 | 0.121 | *<0.001* | −0.774 | 0.086 | *<0.001* |
| treatment | 0.227 | 1.972 | 0.909 | 0.053 | 0.17 | 0.756 | 0.056 | 0.121 | 0.644 |
| sampling date | 14.281 | 0.669 | *<0.001* | 1.12 | 0.058 | *<0.001* | 0.795 | 0.041 | *<0.001* |
| treatment : date | −0.503 | 0.931 | 0.591 | −0.043 | 0.08 | 0.589 | −0.037 | 0.057 | 0.523 |
| random effects | | | | | | | | | |
| $\tau_{00,\,tank}$ | <0.001 | | | <0.001 | | | <0.001 | | |
| $\sigma^2$ | 22.656 | | | 0.168 | | | 0.085 | | |
| $N_{tank}$ | 6 | | | 6 | | | 6 | | |

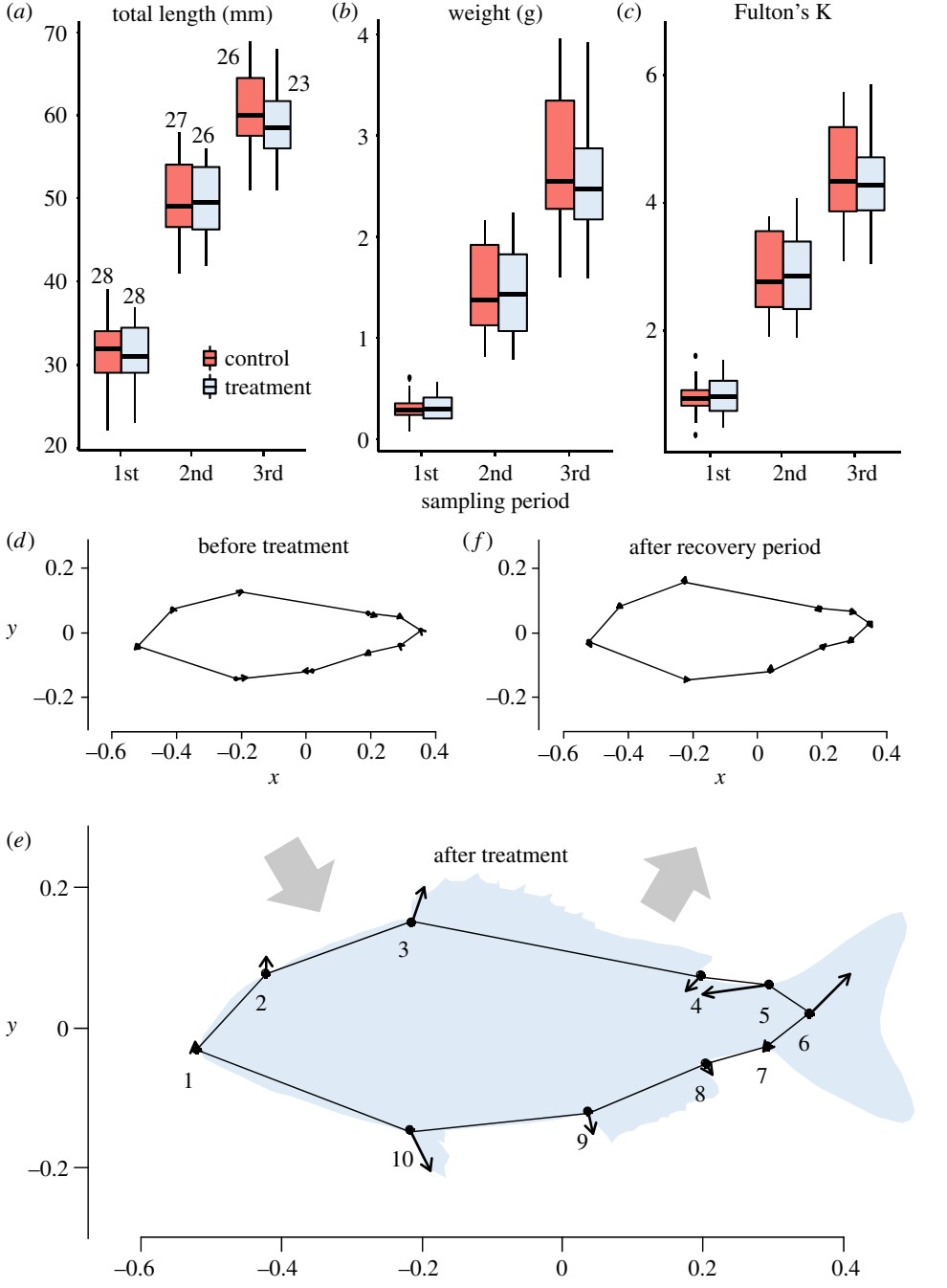

**Figure 1.** Boxplots of *S. aurata* lengths (*a*), eviscerated weight (*b*) and Fulton's K index (*c*) during the experiment. Boxes represent the lower and upper quartiles of the values, the horizontal line the median and the vertical lines the minimum and maximum. The sample size is indicated in (*a*). (*d–f*) Average allometry-corrected body shapes of individuals from the three sampling dates. Vectors show the direction and magnitude (x20) of the landmarks that present differences. (*d*) Before treatment; (*e*) after one month of treatment; (*f*) after the recovery period. Landmark numbers are explained in electronic supplementary material, figure S1. *x* and *y* are the coordinates of the projected landmarks and vectors.

predatory safe state due to gape limitation of predators [4], which may impact the fitness of the individual. A natural next step of our work is, therefore, to investigate the consequences of the PIMDs observed here in the swimming and escape performance in our species case-study or to provide evidence of the fitness consequences of PIMDs.

The reversibility of these PIMDs has received less attention than its expression *per se*. Macroscopic reversible morphometric changes were observed as a response to external environmental stressors in characiform fishes in Argentinian floodplains [17] and microscopically in crucian carp gills [18], both

**Table 2.** Results of the Procrustes MANOVA including the allometry effect on the body shape. A residual randomization permutation procedure is used. $^*p < 0.05$, $^{**}p < 0.01$. Csize is centroid size (see electronic supplementary material). Treatment stands for the effect of 'treatment versus control'.

| sampling | samples | variable | SS | Rsq | F | Pr(>F) |
|---|---|---|---|---|---|---|
| 1st | 56 | ln(Csize) | 0.005878 | 0.16315 | 10.495 | 0.002** |
| | | treatment | 0.000465 | 0.012918 | 0.831 | 0.566 |
| | | residuals | 0.029687 | | | |
| 2nd | 53 | ln(Csize) | 0.0027148 | 0.099932 | 5.7509 | 0.002** |
| | | treatment | 0.0008484 | 0.031228 | 1.7971 | 0.046* |
| | | residuals | 0.0236031 | | | |
| 3rd | 49 | ln(Csize) | 0.0008409 | 0.050429 | 2.4822 | 0.01* |
| | | treatment | 0.0002508 | 0.015039 | 0.7402 | 0.686 |
| | | residuals | 0.0155838 | | | |

as an adaptive response to hypoxia events. However, the reversibility of strong macroscopic morphological changes induced by predatory/conspecific cues has only been described in tadpoles [34] and a freshwater fish [16] and has been suggested for crucian carp [5]. Here we have found that after withdrawal of the stimulus, the induced change disappeared within just a month. Therefore, we provide novel findings on the plastic nature of PIMDs.

PIMDs carry an energetic cost associated with these morphological changes that may only be possible at certain stages or physiological status [35,36]. In our work, juvenile *S. aurata* exposed to predator cues grew at the same rates as control fish revealing an inherent morphological change associated with the energetic demand. Our experiment was performed at the same density and recommended food supply conditions for all the tanks resulting in no differences among treatments in terms of growth. However, in conditions where food is limited, the energetic cost associated with PIMDs may be compensated with a reduction in growth rates [37]. Whether only one trait or both (deeper body and longer tail peduncle) have adaptive value, or whether they are correlated due to metabolic constraints, remain as open questions. On the other hand, morphological changes may only be detectable during particularly plastic stages [16,35,38]. Our main finding suggest reversibility of PIMDs in a marine fish at least at treatment level, although individual reversibility needs to be further addressed by monitoring fish individually. However, the implications of this reversibility, including potential maladaptations later on in their life cycle (e.g. altering reproductive fitness) [1] remains to be tested.

Species-specificity and stocking conditions in cultured fish have profound effects when observing shape changes of fish [39,40]. Notwithstanding the large number of unknowns regarding the consequences of the PIMDs and the reversibility of the phenomenon along development, including how other stressors such as temperature or pollutants may affect PIMDs [2–3], we hypothesize that PIMDs in *S. aurata* may affect individual fitness and also might have a practical application, for example in restocking. Most mortality in released juveniles occurs right after the release event [26], and previous work carried out mostly on salmonid species has shown that learning to identify predator–prey interaction cues and alarm cue stimulus may enhance survival rate *per se* by modifying the prey behaviour [41–43]. Besides the behavioural responses, inducing PIMDs prior to release could also reduce this mortality because (i) deeper-bodied individuals might exhibit better survival in front of gape-limited predators right after release [5], (ii) deeper-bodied individuals may exhibit higher manoeuvrability, increasing escaping opportunities [32], (iii) the longer tail section has been related with more thrust during startle responses [33] and potentially with higher escape responses, and (iv) this is a low-cost technique to be applied in small volumes of water. Further research on PIMDs, their (epi)genetic nature, ecological consequences and applicability is needed to provide insight into the fitness consequences of these morphological responses in *S. aurata*.

Ethics. All experimental procedures were performed by qualified operators at the LIMIA (Balearic Government). Experimental protocols including euthanasia were approved by the competent regional organization (permit no. CEEA 01-11-13). This study did not involve protected or vulnerable species.
Data accessibility. Data files are available in the electronic supplementary material.

Authors' contributions. C.D.-G., J.A., A.G., I.R.-B. and I.A.C. conceived and designed the study and performed the experiments. C.D.-G., M.P. and P.A.-L. analysed the data; C.D.-G., J.A., P.A.-L. and I.A.C. wrote the initial manuscript and all the authors gave their final approval.

Competing interests. Authors declare no competing interests.

Funding. This study was funded by the Spanish Ministry of Economy and Competitiveness (MINECO, grant no. CTM2011-23835). C.D.-G. was funded by a fellowship (FPI-INIA-2012) from the National Institute of Agricultural and Food Research and Technology. P.A.-L. was supported through 'Juan de la Cierva' Post-doc grant (grant nos. IJCI-2016-27681 and IJC-2015-25595) funded by MINECO. J.A. was supported by a Ramón y Cajal Grant funded by the Spanish Ministry of Science, Innovation and Universities (grant no. RYC2018-024488-I). The rest of the coauthors are government employees.

Acknowledgements. This work is a contribution of the Joint Research Unit IMEDEA-LIMIA. We would like to thank Aquicultura Balear S.A.U., CULMAREX for providing the juvenile fish for the study and the personnel from LIMIA for assistance during the experiments at their facilities. We appreciate the revisions from three anonymous reviewers that improved the quality of this manuscript.

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
