## [Reviewer comments · Royal Society Open Science]

Review History

RSOS-191945.R0 (Original submission)

Review form: Reviewer 1

Is the manuscript scientifically sound in its present form?

Yes

Are the interpretations and conclusions justified by the results?

Yes

Is the language acceptable?

Yes

Do you have any ethical concerns with this paper?

No

Have you any concerns about statistical analyses in this paper?

No

Recommendation?

Accept with minor revision (please list in comments)

Comments to the Author(s)

This is an interesting study. The novel contributions are that it is on a marine species and the follow-up measurements on post-exposure demonstrating reversibility of the morphological response.

Comments:

1. line 54 deeper-bodied
2. line 55: mouth gape
3. lines 55-62: Omitted a germane paper by Lonnstedt et al 2013 [nature.com/articles/srep02259](https://doi.org/10.1038/srep02259) that showed - in a marine species - that the size of the false eye spot increases in diameter after exposure to chemical cues of predation
4. line 64: These are alarm cues, not alarm pheromones. A pheromone is a signal that has been shaped over evolutionary time by benefits that accrued to the sender from responses by receivers. A sex pheromone is a signal because senders benefit by increased probability of mating. An alarm cue is released upon injury or death. Receivers benefit from this information but the sender does not. Therefore it is a cue. See refs 11, 32 for detailed explanation of this point.
5. line 209-218: Induced morphological responses may improve survival but so too does learned recognition of predator odor, which would occur simultaneously with a predator odour + alarm cue stimulus. There are several papers, mostly on salmonids, that have worked on predator training of hatchery-reared fish. Those papers should be included in this section of the discussion. A general review, such as ref 32, will list those papers.

Review form: Reviewer 2

Is the manuscript scientifically sound in its present form?

Yes

Are the interpretations and conclusions justified by the results?

No

Is the language acceptable?

Yes

Do you have any ethical concerns with this paper?

No

Have you any concerns about statistical analyses in this paper?

No

Recommendation?

Major revision is needed (please make suggestions in comments)

Comments to the Author(s)

The manuscript RSOS-191945 entitled Reversible morphological changes in a juvenile marine fish after exposure to predatory alarm cues, it is an interesting work leading by Diaz-Gil and co-

workers, focused on phenotypic plasticity phenomena and its implication to predator-prey interactions. By an experimental approach, the authors seeking for a pattern of reversibility of phenotypic plasticity in morphological traits, which are linked with presence alarm cues released by a specific predator into the water where *Sparus aurata* (prey species) was maintained through the experiment. Diaz-Gil et al., from the pattern of variation in morphological shape in individuals exposed to alarm cues vs. control, speculate about the meaning of plasticity in ecological and even in its potential economic implications. The manuscript has been well written, it is easy to follow, and the length is according to the scope of the questions presented. However, there are several theoretical and methodological aspects, which I cannot entirely agree with, or either I did not understand completely. That concern makes me hard to follow the way as the authors interpreted and conclude their results, so I am afraid some deep changes are needed to improve the approach used and address the finding and conclusions adequately.

The authors established an apparent gap of knowledge about mechanisms and implications of phenotypic plasticity (Line 23-24), especially in marine animals. Although I agree that other biological models such as *Daphnia*, *Caenorhabditis*, or *Arabidopsis* have been broad studies instead of wild animals as fishes, there are good amount of literature with Sticklebacks (marine fish) that have studied on phenotypic plasticity, from morphological to behavioral responses and their implications in adaptive evolution (Valimaki et al. 2012). So, I do not think that it is appropriate to establish this issue as a problem to solve, also considering the species-specific approach of the present manuscript. In line with this criticism, in my opinion, the authors defined phenotypic plasticity in a way that may lead to confusion, which may explain the later approach used and the final interpretation of the results.

Inline 43-46 authors defined plasticity as phenotypic responses in front of the environmental variability, which must be linked directly with increment in fitness values and directly connected with adaptation to new environments. Well, this sentence under current literature might be understood as a prediction of the evolution of plasticity instead of a mechanism in action which explains the presence of plasticity in organisms nowadays. Than Scheiner 1993, as Price 2002, established that the evolution of plasticity required environmental signatures to express on time specific plastic responses that should evolve into the species because it is linked with relative fitness. However, the presence of current plastic responses does not necessarily result in current environmental conditions, so phenotypes in several cases do not match with the environmental signatures, leading to interpret those cases as maladaptive responses. Well, to solve this problem, it is being suggested in the literature that a condition to establish a link between phenotypic plasticity and adaptive meaning, it is necessary to establish the biological mechanisms of the induces responses. In other words, to establish the adaptive meaning of changes of body shape in *S. aurata*, it is needed establish if these phenotypic responses are useful to avoid predation. If this antecedent is unknow, the hypothesis between changes in body shape and avoid predation is spurious. This problem is not addressed along with the manuscript, which, in my opinion, is crucial criticisms that must be solved. So authors should, by literature or experimental evidence, establish the relationship between plasticity and the benefit in fitness before to say if this morphological plastic pattern is adaptive or not.

My other main concern is related to the experimental design to test plastic reversibility. The authors set up an experiment where animals were exposed to the presence and absence of alarm cues sequentially. From the total of animal exposed, a sample was taken to assess the phenotypic variation as a consequence of to be in the presence of predation risk. Well, the main problem with this experimental design is if the phenotypic variation in the presence and absence of alarm cues happens at an individual or population level. So, the authors are not able to distinguish if these changes actually are reversible or the presence of alarm cues just change the frequency of the animals with a plastic (fixed) response from each tank. Although we can assume reversible response happens, the design is not appropriate. Authors should individualize (for example 20) and expose them to alarm cues in a different time, at least they would be able to estimate repeatability of the phenotypic responses. So, in my opinion, this design could be used to probe plasticity, but not the reversibility, which a central part of the manuscript.

Line 66-70. What about other stressors such as temperature or pollutants? If the point illustrates the potential other stressors, I suggest including the whole range of stressors or no one.

Line 85-89. The core question of the manuscript implies phenotypic response as a consequence of experience predation risk, but Are these fish predators actually predator of *S. aurata* in the wild? This issue has to be attended, to understand the nature of the phenotypic responses.

Line 191-195. In the absence of a significant difference in the Condition factor (Fulton's K) between the experimental treatment. How can the authors discuss the presence of the cost of a trade-off as a consequence of express phenotypic plasticity? It seems too many speculations. The difference among temporal sampling may just imply the improvement of individual performance as a consequence of the captivity conditions.

Decision letter (RSOS-191945.R0)

18-Feb-2020

Dear Dr Catalan,

The editors assigned to your paper ("Reversible morphological changes in a juvenile marine fish after exposure to predatory alarm cues") have now received comments from reviewers. We would like you to revise your paper in accordance with the referee and Associate Editor suggestions which can be found below (not including confidential reports to the Editor). Please note this decision does not guarantee eventual acceptance.

Please submit a copy of your revised paper before 12-Mar-2020. Please note that the revision deadline will expire at 00.00am on this date. If we do not hear from you within this time then it will be assumed that the paper has been withdrawn. In exceptional circumstances, extensions may be possible if agreed with the Editorial Office in advance. We do not allow multiple rounds of revision so we urge you to make every effort to fully address all of the comments at this stage. If deemed necessary by the Editors, your manuscript will be sent back to one or more of the original reviewers for assessment. If the original reviewers are not available, we may invite new reviewers.

If your study uses humans or animals please include details of the ethical approval received, including the name of the committee that granted approval. For human studies please also detail

whether informed consent was obtained. For field studies on animals please include details of all permissions, licences and/or approvals granted to carry out the fieldwork.

- Data accessibility

If you wish to submit your supporting data or code to Dryad (<http://datadryad.org/>), or modify your current submission to dryad, please use the following link:
<http://datadryad.org/submit?journalID=RSOS&manu=RSOS-191945>

- Competing interests

- Authors' contributions

- Acknowledgements

- Funding statement

Kind regards,

Andrew Dunn

on behalf of Prof Kevin Padian (Subject Editor)
 openscience@royalsociety.org

Associate Editor's comments:

With apologies for the unusual delay in completing review (an unusual number of referees had to be approached to secure these reports), two reviewers have now assessed your paper. Please ensure you tackle the linguistic and other queries of reviewer 1, but perhaps the more problematic queries of reviewer 2 should be prioritised - please ensure that your revision not only includes reasonable adjustments to respond to these concerns but also includes a point-by-point response to them. Good luck and we'll look forward to receiving your revision in due course.

Comments to Author:

Reviewers' Comments to Author:

Reviewer: 1

Comments to the Author(s)

This is an interesting study. The novel contributions are that it is on a marine species and the follow-up measurements on post-exposure demonstrating reversibility of the morphological response.

Comments:

1. line 54 deeper-bodied
2. line 55: mouth gape
3. lines 55-62: Omitted a germane paper by Lonnstedt et al 2013 nature.com/articles/srep02259 that showed - in a marine species - that the size of the false eye spot increases in diameter after exposure to chemical cues of predation
4. line 64: These are alarm cues, not alarm pheromones. A pheromone is a signal that has been shaped over evolutionary time by benefits that accrued to the sender from responses by receivers. A sex pheromone is a signal because senders benefit by increased probability of mating. An alarm cue is released upon injury or death. Receivers benefit from this information but the sender does not. Therefore it is a cue. See refs 11, 32 for detailed explanation of this point.
5. line 209-218: Induced morphological responses may improve survival but so too does learned recognition of predator odor, which would occur simultaneously with a predator odour + alarm cue stimulus. There are several papers, mostly on salmonids, that have worked on predator training of hatchery-reared fish. Those papers should be included in this section of the discussion. A general review, such as ref 32, will list those papers.

Reviewer: 2

Comments to the Author(s)

The manuscript RSOS-191945 entitled Reversible morphological changes in a juvenile marine fish after exposure to predatory alarm cues, it is an interesting work leading by Diaz-Gil and co-workers, focused on phenotypic plasticity phenomena and its implication to predator-prey interactions. By an experimental approach, the authors seeking for a pattern of reversibility of phenotypic plasticity in morphological traits, which are linked with presence alarm cues released by a specific predator into the water where *Sparus aurata* (prey species) was maintained through the experiment. Diaz-Gil et al., from the pattern of variation in morphological shape in

individuals exposed to alarm cues vs. control, speculate about the meaning of plasticity in ecological and even in its potential economic implications. The manuscript has been well written, it is easy to follow, and the length is according to the scope of the questions presented. However, there are several theoretical and methodological aspects, which I cannot entirely agree with, or either I did not understand completely. That concern makes me hard to follow the way as the authors interpreted and conclude their results, so I am afraid some deep changes are needed to improve the approach used and address the finding and conclusions adequately.

The authors established an apparent gap of knowledge about mechanisms and implications of phenotypic plasticity (Line 23-24), especially in marine animals. Although I agree that other biological models such as *Daphnia*, *Caenorhabditis*, or *Arabidopsis* have been broad studies instead of wild animals as fishes, there are good amount of literature with Sticklebacks (marine fish) that have studied on phenotypic plasticity, from morphological to behavioral responses and their implications in adaptive evolution (Valimaki et al. 2012). So, I do not think that it is appropriate to establish this issue as a problem to solve, also considering the species-specific approach of the present manuscript. In line with this criticism, in my opinion, the authors defined phenotypic plasticity in a way that may lead to confusion, which may explain the later approach used and the final interpretation of the results.

Inline 43-46 authors defined plasticity as phenotypic responses in front of the environmental variability, which must be linked directly with increment in fitness values and directly connected with adaptation to new environments. Well, this sentence under current literature might be understood as a prediction of the evolution of plasticity instead of a mechanism in action which explains the presence of plasticity in organisms nowadays. Than Scheiner 1993, as Price 2002, established that the evolution of plasticity required environmental signatures to express on time specific plastic responses that should evolve into the species because it is linked with relative fitness. However, the presence of current plastic responses does not necessarily result in current environmental conditions, so phenotypes in several cases do not match with the environmental signatures, leading to interpret those cases as maladaptive responses. Well, to solve this problem, it is being suggested in the literature that a condition to establish a link between phenotypic plasticity and adaptive meaning, it is necessary to establish the biological mechanisms of the induces responses. In other words, to establish the adaptive meaning of changes of body shape in *S. aurata*, it is needed establish if these phenotypic responses are useful to avoid predation. If this antecedent is unknow, the hypothesis between changes in body shape and avoid predation is spurious. This problem is not addressed along with the manuscript, which, in my opinion, is crucial criticisms that must be solved. So authors should, by literature or experimental evidence, establish the relationship between plasticity and the benefit in fitness before to say if this morphological plastic pattern is adaptive or not.

My other main concern is related to the experimental design to test plastic reversibility. The authors set up an experiment where animals were exposed to the presence and absence of alarm cues sequentially. From the total of animal exposed, a sample was taken to assess the phenotypic variation as a consequence of to be in the presence of predation risk. Well, the main problem with this experimental design is if the phenotypic variation in the presence and absence of alarm cues happens at an individual or population level. So, the authors are not able to distinguish if these changes actually are reversible or the presence of alarm cues just change the frequency of the animals with a plastic (fixed) response from each tank. Although we can assume reversible response happens, the design is not appropriate. Authors should individualize (for example 20) and expose them to alarm cues in a different time, at least they would be able to estimate repeatability of the phenotypic responses. So, in my opinion, this design could be used to probe plasticity, but not the reversibility, which a central part of the manuscript.

Line 66-70. What about other stressors such as temperature or pollutants? If the point illustrates the potential other stressors, I suggest including the whole range of stressors or no one.

Line 85-89. The core question of the manuscript implies phenotypic response as a consequence of

experience predation risk, but Are these fish predators actually predator of *S. aurata* in the wild? This issue has to be attended, to understand the nature of the phenotypic responses.

Line 191-195. In the absence of a significant difference in the Condition factor (Fulton's K) between the experimental treatment. How can the authors discuss the presence of the cost of a trade-off as a consequence of express phenotypic plasticity? It seems too many speculations. The difference among temporal sampling may just imply the improvement of individual performance as a consequence of the captivity conditions.

Author's Response to Decision Letter for (RSOS-191945.R0)

See Appendix A.

RSOS-191945.R1 (Revision)

Review form: Reviewer 2

Is the manuscript scientifically sound in its present form?

Yes

Are the interpretations and conclusions justified by the results?

Yes

Is the language acceptable?

Yes

Do you have any ethical concerns with this paper?

No

Have you any concerns about statistical analyses in this paper?

Recommendation?

Accept with minor revision (please list in comments)

Comments to the Author(s)

This new versión of the manuscript RSOS-191945.R1, it is a definite improvement of the manuscript sent previously. Authors have been addressed most of the suggestions made for the reviewers, doing emphasis on focus the questions and final implications of the findings from the manuscript. The reference update, the amended of the final part of the introduction, and discussion make me much easier to understand the implications of the data presented here. Just a couple of final comments:

Line 24: I would avoid using "innate" mostly because this is not tested ore ven defined in the manuscript. I suggest delete it.

Line 272: check typo

Line 325: check typo

Decision letter (RSOS-191945.R1)

14-Apr-2020

Dear Dr Catalan:

On behalf of the Editors, I am pleased to inform you that your Manuscript RSOS-191945.R1 entitled "Reversible morphological changes in a juvenile marine fish after exposure to predatory alarm cues" has been accepted for publication in Royal Society Open Science subject to minor revision in accordance with the referee suggestions. Please find the referees' comments at the end of this email.

The reviewers and Subject Editor have recommended publication, but also suggest some minor revisions to your manuscript. Therefore, I invite you to respond to the comments and revise your manuscript.

- Ethics statement

- Data accessibility

<http://datadryad.org/submit?journalID=RSOS&manu=RSOS-191945.R1>

- Competing interests

- Authors' contributions

AB carried out the molecular lab work, participated in data analysis, carried out sequence alignments, participated in the design of the study and drafted the manuscript; CD carried out the statistical analyses; EF collected field data; GH conceived of the study, designed the study,

coordinated the study and helped draft the manuscript. All authors gave final approval for publication.

- Acknowledgements

- Funding statement

Because the schedule for publication is very tight, it is a condition of publication that you submit the revised version of your manuscript before 23-Apr-2020. Please note that the revision deadline will expire at 00.00am on this date. If you do not think you will be able to meet this date please let me know immediately.

on behalf of Prof Kevin Padian (Subject Editor)
openscience@royalsociety.org

Associate Editor Comments to Author:

Thank you for taking such care to respond to the reviewer commentary. After seeking further advice, the Editors are pleased to recommend acceptance subject to your completing the remaining (minor) revisions to your manuscript. Thanks for your support of Royal Society Open Science.

Reviewer comments to Author:

Reviewer: 2

Comments to the Author(s)

This new version of the manuscript RSOS-191945.R1, it is a definite improvement of the manuscript sent previously. Authors have been addressed most of the suggestions made for the reviewers, doing emphasis on focus the questions and final implications of the findings from the manuscript. The reference update, the amended of the final part of the introduction, and discussion make me much easier to understand the implications of the data presented here. Just a couple of final comments:

Line 24: I would avoid using “innate” mostly because this is not tested or even defined in the manuscript. I suggest delete it.

Line 272: check typo

Line 325: check typo

Author's Response to Decision Letter for (RSOS-191945.R1)

See Appendix B.

Decision letter (RSOS-191945.R2)

20-Apr-2020

Dear Dr Catalan,

It is a pleasure to accept your manuscript entitled "Reversible morphological changes in a

juvenile marine fish after exposure to predatory alarm cues" in its current form for publication in Royal Society Open Science.

on behalf of Prof Kevin Padian (Subject Editor)
openscience@royalsociety.org

Appendix A

Answer to referees

We thank the referees for their valuable comments. The new version has benefited from essential clarifications derived from their comments.

Comments to Author:

Reviewers' Comments to Author:

Reviewer: 1

Comments to the Author(s)

This is an interesting study. The novel contributions are that it is on a marine species and the follow-up measurements on post-exposure demonstrating reversibility of the morphological response.

Thanks for the comments, we have addressed them properly and we believe they have clarified and polished the final result of the manuscript.

Comments:

1. line 54 deeper-bodied

Done thanks

2. line 55: mouth gape

Indeed

3. lines 55-62: Omitted a germane paper by Lonnstedt et al 2013 [nature.com/articles/srep02259](https://doi.org/10.1098/rsbl.2018.0032) that showed - in a marine species - that the size of the false eye spot increases in diameter after exposure to chemical cues of predation

We are fully aware of the manuscript from Lönnstedt and collaborators and was cited in earlier versions of this manuscript. However the recent situation involving these authors in a rather mediatic case of scientific misconduct (involving a paper in Science <https://science.sciencemag.org/content/352/6290/1213> and a revision of at least one more in Biology Letters <https://royalsocietypublishing.org/doi/10.1098/rsbl.2018.0032>) make us reluctant of citing their work. Moreover, that paper of damselfish eye-spot changes in morphometry did not provide downloadable data to check out the analyses. For these reasons, we prefer not to cite this article.

4. line 64: These are alarm cues, not alarm pheromones. A pheromone is a signal that has been shaped over evolutionary time by benefits that accrued to the sender from responses by receivers. A sex pheromone is a signal because senders benefit by increased probability of mating. An alarm cue is released upon injury or death. Receivers benefit from this information but the sender does not. Therefore it is a cue. See refs 11, 32 for detailed explanation of this point.

Thanks for the comment. This has been changed for clarity (see line 65).

5. line 209-218: Induced morphological responses may improve survival but so too does learned recognition of predator odor, which would occur simultaneously with a

predator odour + alarm cue stimulus. There are several papers, mostly on salmonids, that have worked on predator training of hatchery-reared fish. Those papers should be included in this section of the discussion. A general review, such as ref 32, will list those papers.

We appreciate the reviewer insight. We have included this as an example of other “training tools” that are also affordable and useful in restocking programs (line 215). We think both mechanisms enhance the survival probabilities of the trained individuals, the behavioural conditioned reactions and the purely mechanistic shape-related changes. Mirza and Chivers (2000 and 2002) and Brown et al. (2013) have been included according to the reviewer comment.

Reviewer: 2

Comments to the Author(s)

The manuscript RSOS-191945 entitled Reversible morphological changes in a juvenile marine fish after exposure to predatory alarm cues, it is an interesting work leading by Diaz-Gil and co-workers, focused on phenotypic plasticity phenomena and its implication to predator-prey interactions. By an experimental approach, the authors seeking for a pattern of reversibility of phenotypic plasticity in morphological traits, which are linked with presence alarm cues released by a specific predator into the water where *Sparus aurata* (prey species) was maintained through the experiment. Diaz-Gil et al., from the pattern of variation in morphological shape in individuals exposed to alarm cues vs. control, speculate about the meaning of plasticity in ecological and even in its potential economic implications. The manuscript has been well written, it is easy to follow, and the length is according to the scope of the questions presented. However, there are several theoretical and methodological aspects, which I cannot entirely agree with, or either I did not understand completely. That concern makes me hard to follow the way as the authors interpreted and conclude their results, so I am afraid some deep changes are needed to improve the approach used and address the finding and conclusions adequately.

Thanks for the comments on our research manuscript. We are aware of the limitations of our work highlighted by the referee, and we tried to be very clear about it (see lines 160-169). However, we contend that despite these limitations, our results are highly valuable and encouraging, and help defining future avenues of research with many potential applications. We will try to answer the doubts of the referee in the following sections.

The authors established an apparent gap of knowledge about mechanisms and implications of phenotypic plasticity (Line 23-24), especially in marine animals. Although I agree that other biological models such as *Daphnia*, *Caenorhabditis*, or *Arabidopsis* have been broad studies instead of wild animals as fishes, there are good amount of literature with Sticklebacks (marine fish) that have studied on phenotypic plasticity, from morphological to behavioral responses and their implications in adaptive evolution (Valimaki et al. 2012). So, I do not think that it is appropriate to establish this issue as a problem to solve, also considering the species-specific approach of the present manuscript. In line with this criticism, in my opinion, the authors defined

phenotypic plasticity in a way that may lead to confusion, which may explain the later approach used and the final interpretation of the results.

We understand the point of view of the reviewer. In our initial manuscript we did not intend to state that overall environmental effects have to differ between fish species just because they are marine or freshwater, but it is important to point out that almost all experiments of this kind have been conducted in freshwater fish. Even within the stickleback family, which have been extensively used as model species from the evolutionary point of view with regard to its enormous range of adaptive morphological changes, most of the experimental work has been conducted in freshwater. The paper you cite and other works by Valimaki and collaborators on nine spined stickleback were conducted using several populations of this species, from which only one was native from a saltwater environment (and nonetheless the experiment was conducted in freshwater for all of them). However, it is known that salinity may be one key environmental factors affecting fish plasticity. For example, in another species of sticklebacks (three spined) inhabiting highly different environments (from freshwater to saline), adaptive divergence is revealed although plasticity is reduced in freshwater demes compared to saltwater ones (McCairns and Bernatchez, *Evolution* 64, 1029-1047). According to your comments and our results, we have modified the manuscript to ensure that the claim regarding this point has been put in adequate context. We have included some extra examples of morphological changes on other saltwater species and we have also included the work mentioned by the reviewer (e.g. see lines 57).

Inline 43-46 authors defined plasticity as phenotypic responses in front of the environmental variability, which must be linked directly with increment in fitness values and directly connected with adaptation to new environments. Well, this sentence under current literature might be understood as a prediction of the evolution of plasticity instead of a mechanism in action which explains the presence of plasticity in organisms nowadays. Than Scheiner 1993, as Price 2002, established that the evolution of plasticity required environmental signatures to express on time specific plastic responses that should evolve into the species because it is linked with relative fitness. However, the presence of current plastic responses does not necessarily result in current environmental conditions, so phenotypes in several cases do not match with the environmental signatures, leading to interpret those cases as maladaptive responses. Well, to solve this problem, it is being suggested in the literature that a condition to establish a link between phenotypic plasticity and adaptive meaning, it is necessary to establish the biological mechanisms of the induces responses. In other words, to establish the adaptive meaning of changes of body shape in *S. aurata*, it is needed establish if these phenotypic responses are useful to avoid predation. If this antecedent is unknow, the hypothesis between changes in body shape and avoid predation is spurious. This problem is not addressed along with the manuscript, which, in my opinion, is crucial criticisms that must be solved. So authors should, by literature or experimental evidence, establish the relationship between plasticity and the benefit in fitness before to say if this morphological plastic pattern is adaptive or not.

We agree with the reviewer in this point and we provide literature support for our interpretation of the obtained results with respect to the link between plasticity and fitness in cases similar to ours. For example, literature on PIMDs regarding to changes in the height of the fish and how this trait poses difficulties for predatory handling of the prey do exist (e.g. Nilsson PA, Brönmark C, Pettersson LB. 1995 Benefits of a

predator-induced morphology in crucian carp. *Oecologia* 104, 291–296). In our discussion (now modified), we contemplate that a similar interpretation could be made in our model species (we have revised our claims according to the reviewer comment, see line 176). Of course, we agree that further experiments are needed, and that these potential implications should ideally have been tested in our experimental individuals, using an experimental approach that, for example, measured the escape response in front of a given stimulus (either real or simulated predation). Unfortunately, we did not have at the moment the adequate equipment to test this. We contend that, being this true, our results are still valuable; We now put forward clearly in the Discussion section that our results an interesting first evidence on predatory induced plastic changes in *Sparus aurata* , but whether these changes have an adaptative meaning should be further investigated (see edited text in lines 210-227).

My other main concern is related to the experimental design to test plastic reversibility. The authors set up an experiment where animals were exposed to the presence and absence of alarm cues sequentially. From the total of animal exposed, a sample was taken to assess the phenotypic variation as a consequence of to be in the presence of predation risk. Well, the main problem with this experimental design is if the phenotypic variation in the presence and absence of alarm cues happens at an individual or population level. So, the authors are not able to distinguish if these changes actually are reversible or the presence of alarm cues just change the frequency of the animals with a plastic (fixed) response from each tank. Although we can assume reversible response happens, the design is not appropriate. Authors should individualize (for example 20) and expose them to alarm cues in a different time, at least they would be able to estimate repeatability of the phenotypic responses. So, in my opinion, this design could be used to probe plasticity, but not the reversibility, which a central part of the manuscript.

We agree with the referee's opinion in that, ideally, individuals should be tagged or kept in individual tanks; all of this was a methodological challenge. Hundreds of fish requiring enough individual space (at least 100 l per fish) should have been stocked, controlling for the random factor tank. Since recent research has shown that tagging juvenile fish is feasible using micro RFIDs tags, we should consider expanding this research line using individual monitoring of the morphometric change over ontogeny with and without a predatory stimulus (See Faggion S, Sanchez P, Vandeputte et al.. 2020 Evaluation of a European sea bass (*Dicentrarchus labrax* L. post-larval tagging method with ultra-small RFID tags. *Aquaculture* 520, 734945). In our work, sampling was randomly conducted in the six tanks and no tank effect was observed within treatment and control groups. Therefore, even though we could not monitor individuals, changes did not depend on one single tank but occurred in all tanks and in the same direction. We therefore openly admit in the discussion that the design was not perfect: we assumed that the observed effect on morphology was induced by the treatment and the reversibility too. We now strengthen the idea (see line 209) that we cannot separate the effect on individuals, but that this needs to be done in the future. Despite the correct observation made by the referee, we still think that our results (correctly toned down) are valuable and interesting and may encourage further investigation not only by our but other groups.

Line 66-70. What about other stressors such as temperature or pollutants? If the point illustrates the potential other stressors, I suggest including the whole range of stressors or no one.

We agree with the reviewer that other variables may affect PIMDs. We did not pretend to do a full review of environmental causes of change in morphometry because this is the first work of this type performed in our species. We have decided to include only one example of an environmental stressor with morphological implications, plus the human induced ones (we also included a few examples). In addition, we have mentioned other stressors in the discussion section (see line 215).

Line 85-89. The core question of the manuscript implies phenotypic response as a consequence of experience predation risk, but Are these fish predators actually predator of *S. aurata* in the wild? This issue has to be attended, to understand the nature of the phenotypic responses.

Scorpaena porcus (black scorpionfish) is a highly abundant cryptic predatory species that inhabits the shallow rocky and seagrass coastal habitats along the Mediterranean coast. There are several studies that reflect that *S. porcus* do feed on a wide array of fish species including sparids (although they are not properly identified many times, due to degradation of the stomach content). See for example Rafrafi-Nouira et al. (2016); Arculeo, Froglià and Riggio (1993); Harmelin-Vivien, Kaim-Malka, Ledoyer et al. (1989). After their pelagic larval phase *Sparus aurata* settlers arrive to coastal habitats at a size of around 20 mm which is a perfectly fitting size for Scorpaenidae predators. In any case, the predatory cue was prepared using both predatory smell and conspecific alarm cues and we cannot discard that is the last one the main cue of risk over the predator scent itself.

Rafrafi-Nouira S et al. (2016) Food and feeding habits of black scorpionfish, *Scorpaena porcus* (Osteichthyes: Scorpaenidae) from the northern coast of Tunisia (Central Mediterranean). J. Ichthyol. 56, 107–123. (doi:10.1134/S0032945216010112);

Arculeo M, Froglià C, Riggio S. (1993) Food partitioning between *Serranus scriba* and *Scorpaena porcus* (Perciformes) on the infralittoral ground of the south Tyrrhenian sea. Cybium. 17, 251–258.

Harmelin-Vivien ML, Kaim-Malka RA, Ledoyer M, et al. (1989) Food partitioning among scorpaenid fishes in Mediterranean seagrass beds. J. Fish Biol. 34, 715–734. (doi:10.1111/j.1095-8649.1989.tb03352.x).

Line 191-195. In the absence of a significant difference in the Condition factor (Fulton's K) between the experimental treatment. How can the authors discuss the presence of the cost of a trade-off as a consequence of express phenotypic plasticity? It seems too many speculations. The difference among temporal sampling may just imply the improvement of individual performance as a consequence of the captivity conditions.

We agree with the reviewer on this point too, the experimental design was not intended to evaluate the energetic balance itself. To properly understand the energetic trade-off between growing and changing the shape, a diet-controlled experiment should be performed. Controlling individual intake of food and, at the same time, subjecting them to the predatory stimulus is a better (albeit logistically complicated) design to address

this specific point. We are aware that we did not measure the cost of the PIMDs. The only thing we measured that could inform about the energetic trade-off was the average growth and, in our case, we found no empirical links. This is stated in the manuscript and we have slightly modified it to make it more clear in the Discussion section. “Our experiment was performed at the same density and *recommended* food supply conditions for all the tanks *resulting in no differences among treatments in terms of growth*. However, in conditions where food is limited, the energetic cost associated with PIMDs should be compensated with a reduction in growth rates [37].” Even though the observed growth in terms of weight and length was equivalent for both groups we discussed the possible consequences of PIMDs regarding to the energetic balance as a potentially interesting effect that needs further attention. We have mentioned this reviewer comment along the discussion of the manuscript (see lines 207, 213).

We do thank the referee for pointing towards these key points. We believe that the new version of the document is much more clear now, and that key elements that deserve further research are made evident.

Appendix B

Answer to referees

Associate Editor Comments to Author:

Thank you for taking such care to respond to the reviewer commentary. After seeking further advice, the Editors are pleased to recommend acceptance subject to your completing the remaining (minor) revisions to your manuscript. Thanks for your support of Royal Society Open Science.

We thank the Editor and referees for their comments. We have incorporated their minor revisions as detailed below. Further, we have revised all the requirements (Ethics etc,) as suggested by the last email.

Reviewer comments to Author:

Reviewer: 2

Comments to the Author(s)

This new versión of the manuscript RSOS-191945.R1, it is a definite improvement of the manuscript sent previously. Authors have been addressed most of the suggestions made for the reviewers, doing emphasis on focus the questions and final implications of the findings from the manuscript. The reference update, the amended of the final part of the introduction, and discussion make me much easier to understand the implications of the data presented here. Just a couple of final comments:

Line 24: I would avoid using “innate” mostly because this is not tested ore ven defined in the manuscript. I suggest delete it.

Ok, deleted

Line 272: check typo

Ok, changed

Line 325: check typo

OK, changed